# The Impact of a 14-Day Altitude Training Camp on Olympic-Level Open-Water Swimmers’ Sleep

**DOI:** 10.3390/ijerph19074253

**Published:** 2022-04-02

**Authors:** Jacopo Antonino Vitale, Cristian Ieno, Roberto Baldassarre, Marco Bonifazi, Francesca Vitali, Antonio La Torre, Maria Francesca Piacentini

**Affiliations:** 1IRCCS Istituto Ortopedico Galeazzi, 20161 Milan, Italy; antonio.latorre@unimi.it; 2Department of Movement, Human and Health Sciences, University of Rome Foro Italico, 00135 Rome, Italy; cris.ieno89@gmail.com (C.I.); mariafrancesca.piacentini@uniroma4.it (M.F.P.); 3Department of Physiology and Pharmacology, University of Rome La Sapienza, 00185 Rome, Italy; 4Italian Swimming Federation, 00135 Rome, Italy; roberto.baldassarre@me.com (R.B.); marco.bonifazi@unisi.it (M.B.); 5Department of Medicine, Surgery and Neuroscience, University of Siena, 53100 Siena, Italy; 6Department of Neurosciences, Biomedicine and Movement Sciences, University of Verona, 37124 Verona, Italy; francesca.vitali@univr.it; 7Department of Biomedical Sciences for Health, University of Milan, 20161 Milan, Italy; 8Department of Human Physiology and Sports Medicine, Vrije Universiteit Brussel, 1090 Brussels, Belgium

**Keywords:** actigraphy, athlete, training load, sleep quality, altitude, orthopedics, swimming

## Abstract

Despite the common belief that sleep quality at altitude is poor, the scientific evidence to support this notion is still modest. Therefore, the purpose of the present study was to evaluate possible changes of actigraphy-based and subjective sleep parameters in a group of elite open-water swimmers during a 14-day altitude training camp (ATC) at 1500 m. The study subjects were five Olympic-level open-water swimmers (mean age: 25.0 ± 3.2 years; 3 females and 2 males). All subjects wore a wrist activity monitor and filled a sleep diary for 18 consecutive nights, 4 nights before and 14 nights during ATC. The data were then analyzed at four different time points: before ATC (PRE), the first two days of ATC (T1), and after one (T2) and two weeks of ATC (T3). Training load, assessed as the covered distance (km), session rating of perceived exertion (sRPE), and heart rate (HR), was monitored during the week before and the first and second week of ATC. No significant differences in objective and subjective scores of sleep quality were detected, whereas the sleep onset time (*p* = 0.018; η^2^_p_ = 0.83, large) and sleep offset time (*p* < 0.001; η^2^_p_ = 0.95, large) significantly differed among PRE, T1, T2, and T3: elite athletes started to sleep and woke up ≃ 1 h earlier the first two days of ATC compared to PRE (sleep onset time: *p* = 0.049; sleep offset time: *p* = 0.016). Further, an increase in the training volume during the two weeks of the ATC was observed, with the most time spent in a low-intensity regime and an increase in time spent in a high-intensity regime compared to PRE. Sleep quality was not negatively influenced by a 14-day altitude training camp at 1500 m in a group of Olympic-level elite swimmers despite an increase in perceived exertion during training sessions. Nonetheless, early sleep onset and sleep offset times were observed for the first two nights of ATC: elite athletes started to sleep and woke up ≃ 1 h earlier compared to the baseline nights.

## 1. Introduction

Sleep is a biological process with several bio-physiological functions that are essential for recovery and performance in athletes. Elite athletes often do not reach the recommended night-time sleep duration of 7–9 h per 24 h cycle [1], and it has been shown that acute sleep restriction could negatively impact both sport-specific and athletic performance in different sport disciplines [2,3,4]. Further, sex differences in sleep exist, with women reporting more problems in falling asleep or staying asleep than men, whereas male individuals tend to register more obstructive sleep disorders (e.g., sleep apnea) than female subjects [5]. Despite women reporting more sleep problems than men in the general population [6], in sport settings, male athletes are more-evening oriented than female athletes (who in turn show a larger number of morning types) and have a lower percentage of sleep efficiency and slow-wave sleep than women [7,8]. Sleep disturbances in athletes may be explained by the fact that they are constantly exposed to many stressors that impair sleep, such as high training loads and high-intensity training sessions, early morning training and/or late evening competition, long-haul travel, and high levels of anxiety and stress [9,10,11]. Among these variables, altitude exposure may also play a key role in sleep. Very few data are available on the effect of altitude exposure on athletes’ sleep [12]. Altitude determines several physiological compensatory changes, including the hyperventilatory response to arterial desaturation and an increase in sympathetic activity, with increased heart rate and blood pressure [13,14], which affects both performance and perceived exertion [15]. At high altitudes (>2200 m), athletes may experience insomnia symptoms, restless sleep, and, in some cases, they report a subjective sensation of apnea [16], and it seems that appropriate acclimatization is crucial for injury and illness prevention. It has been shown that U20 football players reduced total sleep duration in the first few days after traveling from sea level to 1600 m [17]. Roach et al. [18] observed a reduction in both sleep quality and duration the first night at high altitude (3600 m) in a U17 soccer team and, in addition, sleep quality returned to baseline values after the first week at altitude, whereas sleep quantity did not [18]. Conversely, no changes in sleep duration and quality were detected in runners undertaking six nights of normobaric altitude (2000 m) exposure [19]. Similar to this study, Saw et al. [20] observed that sleep duration was stable and sleep quality improved week-to-week during training camps at moderate altitudes in elite cyclists and swimmers.

Despite the common belief that sleep quality at altitude is poor, the scientific evidence to support this notion is still modest, and, to the best of our knowledge, few studies have yet examined sleep behavior in elite athletes in moderate-altitude training camps (ATCs). Therefore, the purpose of the present study was to evaluate possible changes of actigraphy-based and subjective sleep parameters in a group of elite swimmers, competing at the Olympic level, during a 14-day ATC at 1500 m. Based on the existing evidence, we hypothesized no significant changes in athletes’ sleep parameters.

## 2. Materials and Methods

### 2.1. Subjects and Study Design

The study subjects were five open-water swimmers (Tier 4–5) [21]) specialized in distances between 5 and 25 km (mean age: 25.0 ± 3.2 years; weight: 67.5 ± 12.5 km; height: 167 ± 6.68 cm; 3 females and 2 males; training volume ~80 km per week). Inclusion criteria were age > 18 years and being a world-class swimmer with at least 8 years of experience. Exclusion criteria were tobacco use and use of melatonin and/or medications. The ATC period consisted of 14 days of training performed at 1525 m above sea level (Cervinia-Valtournanche, Aosta, Italy). Training sessions were performed in a 50 m swimming pool, and athletes performed a maximum of 2 swimming training sessions of ~15 km per day. In detail, athletes trained according to the following schedule: from Monday to Friday, in the morning at 08:30–12:00 (swim and gym), and in the afternoon at 16:00–18:30 (swim); on Saturday, only in the morning, at 08:30–11:00 (swim); and Sunday was the day off. Subjects were requested to abstain from extra routine training sessions during the study period to maintain their regular lifestyle and sleep–wake cycle. Before entering the study, all participants gave their written informed consent and received an explanation of the purpose, methods, risks, and benefits of the experimental protocol. The study protocol was approved by the institutional ethics review committee (CARD 1018/13) in compliance with current national and international laws and regulations governing the use of human subjects (Declaration of Helsinki II). This study is a case series, and the experimental procedures were conducted in June 2019 during the in-season period. All subjects wore a wrist activity monitor and filled a sleep diary for 18 consecutive nights, and the data were then analyzed at four different time points: before ATC (PRE), the first two days of ATC (T1), and after one week (T2) and two weeks of ATC (T3).

### 2.2. Chronotype and PSQI

Circadian typology (i.e., the chronotype) was assessed via the Horne–Ostberg Morningness–Eveningness Questionnaire (MEQ) [22]. Participants were classified as morning (M-) type (score ≥ 59), evening (E-) type (score ≤ 41), or neither (N-) type (score 42–58). The Pittsburgh Sleep Quality Index (PSQI) was used to evaluate subjects’ subjective sleep quality and to assess possible sleep complaints [23].

### 2.3. Sleep Parameters

Sleep parameters were monitored with an Actiwatch 2 actigraph (Philips Respironics, Bend, OR, USA) for 18 nights: 4 nights before ATC and 14 nights during the ATC. Sleep parameters were calculated and averaged as follows: four baseline nights for PRE, the first two nights of ATC for T1, the sixth and seventh nights of ATC for T2, and the thirteenth and fourteenth nights of ATC for T3.

A low actigraphic sensitivity threshold (80 counts per epoch) was selected since this cut-off value provides the best combination of sensitivity and specificity in a population of elite athletes [24]. In detail, eight sleep parameters were measured: (1) sleep onset (SOn), the time at which a participant first fell asleep after going to bed; (2) sleep offset (SOff), the time at which a participant last woke before getting up; (3) sleep efficiency (SE), the percentage of time in bed actually spent sleeping; (4) sleep latency (SL), the period of time between bedtime and sleep onset time; (5) wake after sleep onset (WASO), the amount of time spent awake after sleep has been initiated; (6) total sleep time (TST), the number of minutes of sleep obtained during a sleep period; (7) immobility time (IT), the total time, expressed in percentage, spent without recording any movement during time in bed; and (8) fragmentation index (FI), the sum of the percentages of mobility and immobility accesses in 1 min divided by the number of immobility accesses. Together with the actigraph wrist device, each participant received a sleep diary to record bedtime, wake-up time, and the number of nocturnal awakenings. Furthermore, the participant’s self-rating of sleep quality was measured with a 10-point Likert scale (from 0 = very poor sleep quality to 10 = optimal sleep quality).

### 2.4. Training Load

Training load was monitored during the week before ATC (PRE) and the first (T2) and the second (T3) week of the altitude training camp. Heart rate (HR) was recorded continuously (Garmin Forerunner 935; Olathe, KS, USA) during each training session. To define HR intensity zones, individual HR thresholds were previously established from their routine incremental swimming step tests [25]. In addition, athletes were asked to report their session rating of perceived exertion (sRPE) [26] after 30 min of the end of every training session.

### 2.5. Statistical Analysis

Data are reported as mean ± SD. Each objective and subjective sleep parameter was calculated 4 times (PRE, T1, T2, and T3), while training load parameters were calculated 3 times (PRE, T2, and T3) for all subjects and then checked for normality with the Shapiro–Wilk test. All parameters showed normal distribution with the exception of SE, SOff, and training load data. To detect possible differences in sleep among PRE, T1, T2, and T3; and in training load among PRE, T2, and T3, a repeated-measures analysis of variance (RM-ANOVA) followed by the Bonferroni post hoc test, or the equivalent non-parametric Friedman test followed by the Dunn’s multiple comparisons for non-normally distributed variables, was performed. Partial eta-squared (η^2^_p_) was used to determine the magnitude of the effect for significant outcomes (α = 0.05) in the analysis of variance using the small (<0.13), medium (0.13–0.25), and large (>0.25) interpretation for effect size [27]. A *p*-value ≤ 0.05 was considered statistically significant.

## 3. Results

### 3.1. Sleep Parameters

The athletes were three N-types and two M-types (mean MEQ score: 56.6 ± 6.5) and registered a mean PSQI score of 5.2 ± 1.9, with three subjects highlighting a score ≥ 5.

Figure 1 shows the Whiskers plots with single data of all sleep parameters at PRE, T1, T2, and T3, whereas Table 1 displays mean ± SD, *p*-values, and ES. The RM-ANOVA highlighted no significant differences in SE, SL, TST, IT, FI, WASO, and subjective scores of sleep quality, whereas SOn (*p* = 0.018; η^2^_p_ = 0.83, large) and SOff (*p* < 0.001; η^2^_p_ = 0.95, large) significantly differed among PRE, T1, T2, and T3: elite athletes started to sleep and woke up ≃ 1 h earlier the first two days of ATC compared to PRE (SOn: *p* = 0.049; SOff: *p* = 0.016).

### 3.2. Training Data Heart Rate and sRPE Training Load Distribution

Athletes trained for 5 days (8 sessions) in the week before the ATC and 6 days per week during the ATC. Figure 2 shows the training load distribution using volume and HR. Appendix A shows the daily training volume, including sRPE in PRE and during the 2 weeks of the training camp. Training volume shows an increase throughout the training camp (53.52 ± 1.07 in PRE, 82.40 ± 0.00 in T2, and 76.76 ± 0.13 km/week in T3, respectively. No differences were observed in training volume in Z1 between the three periods (*p* = 0.165), while significant differences were found in Z2 and in Z3 (*p* = 0.007; *p* = 0.015). HR shows a significant difference in Z1 between PRE and T3 (*p* = 0.046) and T2 and T3 (*p* = 0.034) and in Z2 between the three periods (*p* < 0.05).

## 4. Discussion

To our knowledge, this is the first study that investigated the changes in sleep parameters in a group of Olympic swimmers during a 14-day altitude training camp at 1500 m. We observed early sleep onset and sleep offset times for the first two nights of ATC; in detail, elite athletes started to sleep and woke up ≃ 1 h earlier compared to the baseline nights. Conversely, the other actigraphy-based sleep parameters and subjective sleep quality did not display any significant changes during the study nights. Our initial hypotheses were partially confirmed.

In the present study, elite swimmers met the National Sleep Foundation (NSF) recommendations for sleep efficiency (i.e., >85%) and WASO (i.e., <40 min) [28]; however, they did not attain the NSF recommended levels for adults of 7–9 h of sleep per 24 h cycle (1): sleep duration was indeed <7 h per night from PRE to T3, with the exception of T2, where the total sleep time was 438.0 min. These results are in line with previous studies highlighting that athletes sleep well below the recommended quantity per night [29]. In addition, athletes in individual sports (e.g., swimming, cycling, or triathlon) tend to go to bed earlier, to wake up earlier, and sleep less overall than athletes from team sports (soccer, volleyball, or basketball) [29,30,31]. In this study, we observed that swimmers went to bed at 21:45 and woke up at 05:03 on the first two days of the ATC, significantly earlier than the baseline values (22:48 and 05:54, respectively). The results can be explained by the athletes’ training schedules: during the first days of the altitude training camp, swimmers were forced to wake up early due to the morning training sessions. Early morning training sessions are indeed able to reduce total sleep time in individual athletes, as confirmed by Sargent et al. [32], who showed that training sessions at 06:00 h severely restrict the amount of sleep obtained by world-class swimmers during a training camp.

Nonetheless, in the present study, sleep duration and quality did not significantly change across the study nights, highlighting that altitude did not impact these sleep parameters. We did not detect any significant change during the altitude training camp: SE, SL, IT, FI, WASO, and subjective scores of sleep quality were consistent in all the evaluation time points. This result is in line with the existing literature showing that athletes’ sleep at low altitudes (<1600 m) is not typically impaired [33] and that sleep disturbances (e.g., a reduction in deep sleep and an impairment of sleep continuity) tend to occur only at altitudes above 2000 m in the initial days after exposure [34]. Lastella et al. observed a shorter TST but no effects on sleep quality in the days after traveling from sea level to 1600 m in football players [17] and, similarly, Saw and colleagues [20] observed that sleep duration remained stable during moderate ATC in a sample of elite cyclists and swimmers. Furthermore, Hoshikawa et al. [19] reported no changes in sleep duration and SE in runners undertaking six nights of normobaric altitude (2000 m) exposure.

To our knowledge, only one study investigated training load distribution during an ATC in elite open-water swimmers [35]. Similarly, the authors reported an increase in training volume during the first two weeks of the ATC, with the majority of training time spent at low intensities (Z1) and, compared to PRE, a higher amount of time spent in Z3. These results seem to confirm that elite athletes are able to increase training intensity after a few days in altitude, as already shown by Pugliese et al. [36] in an elite race walker and marathon runner. The fact that the internal training load does not always exactly reflect the external training load has already been reported in open-water swimmers [35], but specifically, during altitude training, coaches need to be aware that the perception of effort may shift towards higher-intensity zones and therefore carefully monitor their athletes to avoid nonfunctional overreaching [35].

Some limitations need to be acknowledged. First, the sample size was low, limiting the significance of the results, even if all athletes were elite, Olympic-level athletes. Second, the athletes of this study were recruited from a single discipline (i.e., swimming) and the results may not be generalizable to other sport disciplines [29,30,31]. Third, no inter-chronotype differences were investigated due to the small sample size of each sub-group. Fourth, athletes’ daytime napping behavior was not evaluated, and naps may represent a valid strategy for athletes to recover after nights of sleep restriction [37]. Fifth, pooling the sleep data of the four nights before the ATC (including the day of travel) introduced a potential confounding factor and, consequently, a real baseline sleep value was not obtained by the participants. Lastly, this was a field-based ecological study; therefore, some factors (such as the room temperature, light exposure, and caffeine or alcohol intake) were not controlled and may have affected the athletes’ sleep/wake behavior.

## 5. Conclusions

Sleep was not negatively influenced by a 14-day altitude training camp at 1500 m in a group of Olympic-level elite swimmers despite an increase in perceived exertion during training sessions. Further, subjective sleep quality was similar at sea level and at low-altitude exposure, whereas sleep timing (i.e., sleep onset and offset) was influenced by early morning training sessions during the training camp. Given that the amount of sleep an elite athlete obtains is dictated by the training schedule and that acute or chronic sleep restriction may influence different components of the physical performance in athletes [3,4], it is advisable, where possible, to schedule morning training sessions at a time when athletes have the possibility to prepare for training and wake up after sunrise. Providing sleep hygiene education meetings on minimizing phone, computer, and video use prior to sleep can aid in achieving earlier bedtimes for athletes [38,39]. However, further scientific data are required before generalizing on the limited impact of altitude on athletes’ sleep and, in addition, the study of athletes’ sleep in the recovery phase following altitude training camps, when altitude-induced adaptations occur, may represent a possible area of interest for future studies.

## Figures and Tables

**Figure 1 ijerph-19-04253-f001:**
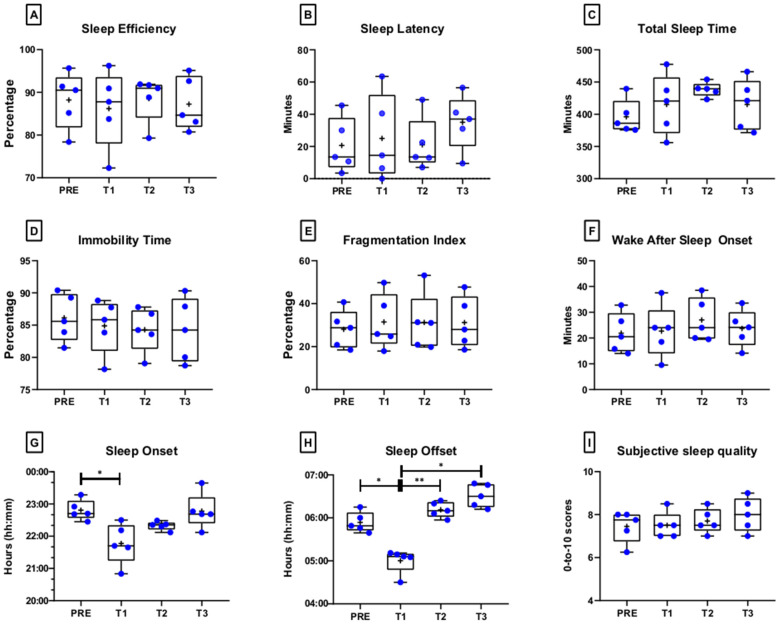
Whisker plot with single data (blue circles) of median, first, and third quartiles and minimum and maximum values of actigraphy-based (panels **A**–**H**) and subjective (panel **I**) sleep parameters before ATC (PRE), the first two days of ATC (T1), and after one week (T2) and two weeks of ATC (T3). *: *p* < 0.05; **: *p* < 0.01.

**Figure 2 ijerph-19-04253-f002:**
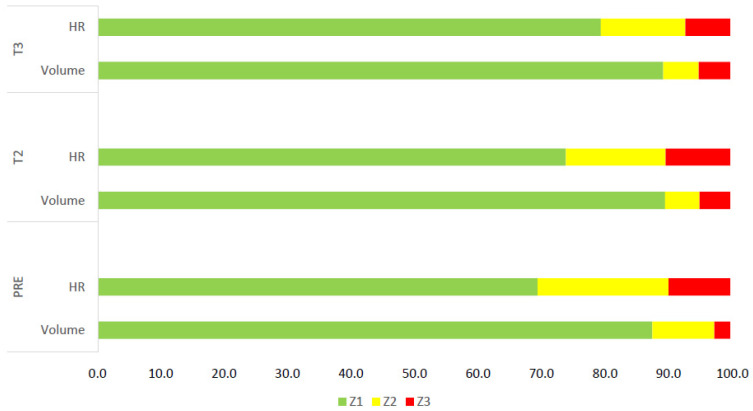
Training intensity distribution using training volume, expressed in km, and HR during PRE, T2, and T3. All values are expressed as % of training/time sessions. HR, heart rate.

**Table 1 ijerph-19-04253-t001:** Actigraphy-based sleep parameters and subjective sleep quality at PRE, T1, T2, and T3. Comparison among PRE, T1, T2, and T3 of sleep parameters. The data are reported as mean ± SD. *: SE and SOff were not normally distributed; therefore, the equivalent non-parametric Friedman test followed by Dunn’s multiple comparisons was utilized.

	PRE	T1	T2	T3	RM-ANOVA	Partial Eta-Squared	Bonferroni Post Hoc Test
SE (%) *	88.2 ± 6.6	86.2 ± 9.0	88.5 ± 5.3	87.2 ± 6.2	*p* = 0.561	-	-
SL (mis)	20.7 ± 16.9	25.0 ± 26.5	21.0 ± 16.6	35.0 ± 17.1	*p* = 0.311	-	-
TST (mim)	396.0 ± 24.6	415.0 ± 46.0	438.0 ± 11.1	415.0 ± 39.5	*p* = 0.269	-	-
IT (%)	86.1 ± 3.7	84.9 ± 4.2	84.3 ± 3.4	84.2 ± 4.9	*p* = 0.667	-	-
FI (%)	28.1 ± 4.9	31.5 ± 12.8	31.3 ± 13.2	31.2 ± 12.0	*p* = 0.333	-	-
WASO (min)	21.9 ± 7.7	22.7 ± 10.2	27.0 ± 8.4	23.7 ± 7.2	*p* = 0.294	-	-
SOn (hh:mm)	22:48 ± 0:18	21:45 ± 0:38	22:19 ± 0:08	22:59 ± 0:33	*p* = 0.018	F_3,4_ = 6.947 η^2^_p_ = 0.83, large	PRE > T1 (*p* = 0.049)
SOff (hh:mm) *	05:54 ± 0:14	05:03 ± 0:17	06:11 ± 0:11	06:31 ± 0:16	*p* < 0.001	F_2,88_ = 29.39 η^2^_p_ = 0.95, large	PRE > T1 (*p* = 0.016); T2 > T1 (*p* = 0.006); T3 > T1 (*p* = 0.010)
Subjective SQ (0-to-10 score)	7.4 ± 0.7	7.5 ± 0.6	7.7 ± 0.6	8.0 ± 0.8	*p* = 0.493	-	-

Abbreviations: RM-ANOVA: repeated-measures analysis of variance; SE: sleep efficiency; SL: sleep latency; TST: total sleep time; IT: immobility time; FI: Fragmentation Index; WASO: wake after sleep onset; SOn: sleep onset; SOff: sleep offset; SQ: sleep quality.

## Data Availability

Data will be shared upon specific request.

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
