# Peer review of "The Impact of a 14-Day Altitude Training Camp on Olympic-Level Open-Water Swimmers’ Sleep"

_ijerph, 2022, doi:10.3390/ijerph19074253_

Round 1

Reviewer 1 Report

The aim of the present study was to assess objective and subjective sleep parameters in a group of five elite Olympic swimmers during a 14-day altitude training camp at 1525m. Main results showed that sleep onset time and sleep-offset time only significantly differed among pre, T1, T2 and T3: elite athletes started to sleep and woke up ≃ 1hr earlier the first two days of the altitude training camp compared to pre. I have the following comments to address to the authors.

As mentioned by the authors in the discussion, the change in sleep onset time and sleep-offset time seems more related to the training schedule than the altitude training-induced stress per se. I would suggest mentioning this “training schedule effect” on elite athletes’ sleep from the introduction. Additionally, we currently do not know if the training time was consistent throughout the camp. From my point of view, it is important to present into the methodology the training schedule - morning and afternoon, rest - throughout the camp in detail.

In the methodology, the authors mention that four baseline nights were included for PRE (which may be insufficient to establish ‘true’ baseline values), the sixth and seventh nights of the camp were used for T2, and the thirteenth and fourteenth nights of the camp were used for T3. Several studies have showed that the variability (both inter- and intra-) of sleep variables is high. In this respect, I do not understand why the authors did not follow sleep throughout the entirety of the nights. From my point of view, this is a serious limitation of the present work that should be presented as so. Additionally, the sleep achieved during napping was not assessed whereas studies (e.g. Sargent et al.) have showed that napping is a first choice strategy for swimmers to catch up the nocturnal sleep they miss. Finally, I would suggest discussing the potential interest of assessing sleep after the training camp, in the recovery phase, when several altitude-induced adaptations occur.

I would also suggest presenting the travel and potential associated travel fatigue for swimmers to join the altitude training camp. Additionally, there is no information about the sleeping environment of the swimmers, both at baseline and during the camp.

Finally, as mentioned by the authors, the present training camp is at a moderate altitude (1500m) whereas other studies have assessed sleep at higher altitudes (e.g. >2200 meters; 1600m). In this respect, I would suggest to mention in the discussion that future studies are required before generalizing on the limited impact of altitude on athletes’ sleep. One additional paper that may be relevant here: Saugy et al. Sleep Disordered Breathing During Live High-Train Low in Normobaric Versus Hypobaric Hypoxia. High Alt Med & Biology 2016.

Specific comments

To the best of my knowledge, the recommended sleep duration of 7-9 h in the general population is per 24-hour cycle (and not per night). Please amend in both the introduction and the discussion.

Abstract: “the first two weeks of the ATC”. There is a mistake here if the duration of the training camp was two weeks.

Line 91: cit. Please amend.

A low actigraphic sensitivity threshold (80 counts per epoch) was selected. Please justify.

The  presentation of the training load assessment performed during the protocol is very confusing from my point of view.

Author Response

Reviewer 1

The aim of the present study was to assess objective and subjective sleep parameters in a group of five elite Olympic swimmers during a 14-day altitude training camp at 1525m. Main results showed that sleep onset time and sleep-offset time only significantly differed among pre, T1, T2 and T3: elite athletes started to sleep and woke up ≃ 1hr earlier the first two days of the altitude training camp compared to pre. I have the following comments to address to the authors.

A: Thank you for your comments. We worked to improve clarity and quality of our paper.

As mentioned by the authors in the discussion, the change in sleep onset time and sleep-offset time seems more related to the training schedule than the altitude training-induced stress per se. I would suggest mentioning this “training schedule effect” on elite athletes’ sleep from the introduction. Additionally, we currently do not know if the training time was consistent throughout the camp. From my point of view, it is important to present into the methodology the training schedule - morning and afternoon, rest - throughout the camp in detail.

A: We thank the reviewer and agree with this comment. We included in the methods section the training schedule adopted by the swimmers during ATC. Further, in the introduction, we mentioned that both “early morning training and/or late evening competition” are factors able to affect athletes’ sleep.

In the methodology, the authors mention that four baseline nights were included for PRE (which may be insufficient to establish ‘true’ baseline values), the sixth and seventh nights of the camp were used for T2, and the thirteenth and fourteenth nights of the camp were used for T3. Several studies have showed that the variability (both inter- and intra-) of sleep variables is high. In this respect, I do not understand why the authors did not follow sleep throughout the entirety of the nights. From my point of view, this is a serious limitation of the present work that should be presented as so.

A: We thank the reviewer for the comment and agree on his/her comments. Regarding baseline data, we recognized that we do not have “true” baseline sleep values. This was mentioned as a study limitation at the end of the discussion paragraph. In the text: “…, pooling the sleep data of the four nights before the ATC (including the day of travel) introduced a potential confounding factor and, consequently, a real baseline sleep value was not obtained by the participants”.

We have monitored sleep every single night but decided to report data only from the nights that  were in correspondence to the end of the single training blocks.

Additionally, the sleep achieved during napping was not assessed whereas studies (e.g. Sargent et al.) have showed that napping is a first choice strategy for swimmers to catch up the nocturnal sleep they miss. Finally, I would suggest discussing the potential interest of assessing sleep after the training camp, in the recovery phase, when several altitude-induced adaptations occur.

A: We agree and recognize that the lack of information on napping is a study limitation. We mentioned in the text that “…, athletes’ daytime napping behavior was not evaluated and naps may represent a valid strategy for athletes to recover after nights of sleep restriction”.

As suggested, we mentioned in the conclusions the possible interest of studying sleep in the recovery phase after training camps.

I would also suggest presenting the travel and potential associated travel fatigue for swimmers to join the altitude training camp. Additionally, there is no information about the sleeping environment of the swimmers, both at baseline and during the camp.

A: We would like to thank the Reviewer for this comment. The training camp was reached through a 4-hrs train travel and thereafter 1h50 by car. We did not observe any effect of travel fatigue on the RPE reported by the athletes during their first training session; however, we did not collect specific data on their travel fatigue.  Regarding the sleeping environment, although different from their habitual one, it did not seem to impact on overall sleep because the athletes had been in the same facility multiple times over their career. For this reason we believe that the experience of athletes with altitude training and with the facilities is useful in the adaptation

Finally, as mentioned by the authors, the present training camp is at a moderate altitude (1500m) whereas other studies have assessed sleep at higher altitudes (e.g. >2200 meters; 1600m). In this respect, I would suggest to mention in the discussion that future studies are required before generalizing on the limited impact of altitude on athletes’ sleep. One additional paper that may be relevant here: Saugy et al. Sleep Disordered Breathing During Live High-Train Low in Normobaric Versus Hypobaric Hypoxia. High Alt Med & Biology 2016.

A: Thank you for this comment. We added a sentence on the need of further studies at the end of our manuscript, as suggested. We read with great interest the paper by Saugy et al., thank you for this citation. However, we prefer not to directly discuss papers on “live high-train low” approach since we feel that it is beyond our study aims.

Specific comments

To the best of my knowledge, the recommended sleep duration of 7-9 h in the general population is per 24-hour cycle (and not per night). Please amend in both the introduction and the discussion.

A: Amended as suggested.

Abstract: “the first two weeks of the ATC”. There is a mistake here if the duration of the training camp was two weeks.

A: Thanks for reporting, this was modified accordingly.

Line 91: cit. Please amend.

A: amended as suggested, thank you.

A low actigraphic sensitivity threshold (80 counts per epoch) was selected. Please justify.

A: A low actigraphic sensitivity threshold (80 counts per epoch) was selected since this cut-off value provides the best combination of sensitivity and specificity in a population of elite athletes. This was supported by the paper by Sargent, Lastella, Halson, & Roach, 2016. This information was added in the text.

The  presentation of the training load assessment performed during the protocol is very confusing from my point of view.

A: We thank the reviewer for the comment. Figure 2 presents  data only as external (volume) and internal (HR) data. We took out the sRPE data from this figure because it is beyond the scope of this paper to discuss possible differences in monitoring methods between a session in goal approach (RPE) and a time in zone approach (HR). However, we included sRPE data in a new table (presented as supplementary file) as requested by reviewer 3 including daily volume and reported RPE 30 minutes post training.

Reviewer 2 Report

Comments and Suggestions for Authors

This study describes the effect of altitude training on the quality and quantity of athletes' sleep. As the authors' comment, Olympic-level swimmers were the strong point in the present study, but their small number and different sex, even the medium altitude of the training place were the study's weaknesses.

Introduction

The authors should provide some information on possible differences between sexes in sleep behavior.

Statistical analysis, Results, and Discussion

Did the authors feel the need to present some useful correlations of the sleep parameters reported with HR or RPE or somehow with training performance?

Conclusions

p. 259 Furthermore, Hoshikawa et al (15)

The number is rather redundant

p. 291 "it is advisable, where possible, to avoid early morning training session."

this proposal sounds like this training session should be avoided altogether.  I think the authors should change the sentence to a positive one, such as "morning training sessions should be held at a time when athletes have time to prepare for training and wake up after sunrise, or something similar. 

Author Response

Reviewer 2

Comments and Suggestions for Authors

This study describes the effect of altitude training on the quality and quantity of athletes' sleep. As the authors' comment, Olympic-level swimmers were the strong point in the present study, but their small number and different sex, even the medium altitude of the training place were the study's weaknesses.

A: Thank you for the overall positive feedback. We also recognize the study limitations as mentioned by the Reviewer.

Introduction

The authors should provide some information on possible differences between sexes in sleep behavior.

A: Thank you for the suggestion. We mentioned sex differences in sleep for the general population and for athletes in the introduction citing the papers by Gupta (2017), Silva (2019) and Emsellem (2005).

Statistical analysis, Results, and Discussion

Did the authors feel the need to present some useful correlations of the sleep parameters reported with HR or RPE or somehow with training performance?

A: No, the purpose of presenting training data (and training response) was to verify their response to altitude training as previously reported (  Baldassarre et al. 2019)   We took out the RPE from the Figure and included it in a new table (presented as supplementary file) as requested by reviewer 3 including daily volume and reported RPE 30 minutes post training.

Conclusions

  1. 259 Furthermore, Hoshikawa et al (15). The number is rather redundant

A: Thank you, we amended this accordingly.

  1. 291 "it is advisable, where possible, to avoid early morning training session." This proposal sounds like this training session should be avoided altogether.  I think the authors should change the sentence to a positive one, such as "morning training sessions should be held at a time when athletes have time to prepare for training and wake up after sunrise, or something similar. 

A: We agree with this comment. We modified the text accordingly.

Reviewer 3 Report

Title : based on the data shared in the manuscript, it could be useful to rename the title mentioning you are talking about open-water swimmers.

L19 : open-water swimmers

L23 : start with a new sentence here (a dot after ATC).

L32-34 : maybe this sentence deserves to be rewritten because the data showed no impact on sleep quality (important to mention quality), but the schedule of training camp induced a reduction in sleep quantity. And maybe you can mention this fact as youd id in the conclusion part. You decide.

You can add « swimming » in the keywords ?

L46-47 : i understand the sentence but it is no really well written. Please reword.

L50-59 : there is no reference about sleep of swimmers in altitude ? I am sure it does.

L60-62 : i am not a specialist on sleep assessment in altitude, but I am pretty sure you can find some references on literature. Sometimes, sleep was not the main topic of these articles but some data are provided.

L69-71 : just for my information, why you can argue that they are olympic level open-water swimmers ? Not really important to state that on the manuscript but it is important to mention us how you can say that.

L100-101 : does the fact that the measurements were taken at the beginning (T1) and then at the end of the week (T2 & T3) has an impact of the results ?

L126-127 : could you explain why RPE value of 4 is included in Z2 whereas itw as included in Z1 on this article :

Baldassarre R, Bonifazi M, Meeusen R, Piacentini MF. The road to Rio: A brief report of training-load distribution of open-382 water swimmers during the Olympic season. Int J Sports Physiol Perform. 2019;14(2).

L142 : all the results are comparing differences in mean groups (with just 5 subjects). Do you think it is relevant to share some individual changes, or at least to mention if there was individual differences throughout the periods ? Please explain here and on the manuscript.

L212-221 : it was not possible to calculate a daily training volume/load in order to also compare data with T1 ? I think the reader to deserve more detailed data for that period.

Figure 2 is clear but could be really improved in terms of graphics. Please delete the grid pattern, the borders. Maybe color could be green for Z1, orange for Z2, and red for Z3.

L235-251 : the paragraph is great here but maybe you can shorten some general ideas and add one or two comments about the non-impact on sleep quality/quantity by altitude.

Is that possible that we did not observe negative impact because of the general poor sleep quantity ?

L261 : I think another comment on altitude exposure could be great. I mean the fact that 1500m is moderate altitude and allows to sleep well, but maybe it doesn’t allow sufficient physiological changes ?

L262-272 : I think in this paragraph, two points are missing.

The first is some comparison with training intensity distribution in other elite athletes or swimmers. A reference on the polarized training concept could be useful.

The second – in line with the previous one – could be to discuss the differences observed on intensity distribution depending on physiological measurements (HR, RPE, volume).

L274-276 : could you tell why swimmers are different ? Or not ?

Author Response

Reviewer 3

Title : based on the data shared in the manuscript, it could be useful to rename the title mentioning you are talking about open-water swimmers.

A: Thank you for this suggestion. We modified the title accordingly.

L19 : open-water swimmers

A: Amended as suggested.

L23 : start with a new sentence here (a dot after ATC).

A: Amended as suggested.

L32-34 : maybe this sentence deserves to be rewritten because the data showed no impact on sleep quality (important to mention quality), but the schedule of training camp induced a reduction in sleep quantity. And maybe you can mention this fact as you did in the conclusion part. You decide.

A: We got this point by the Reviewer. We agree with this comment and the text was modified accordingly. We mentioned “sleep quality” and “sleep onset and offset times” since sleep quantity was not significantly influenced.

You can add « swimming » in the keywords ?

A: yes, this is a nice idea for possible future searchers. Thank you.

L46-47 : i understand the sentence but it is no really well written. Please reword.

A: We reworded this sentence as suggested, thank you!

L50-59 : there is no reference about sleep of swimmers in altitude ? I am sure it does.

A: The Reviewer is right, we mentioned and discussed the paper by Saw et al., 2018

L60-62 : i am not a specialist on sleep assessment in altitude, but I am pretty sure you can find some references on literature. Sometimes, sleep was not the main topic of these articles but some data are provided.

A: Yes, the reviewers is right. We cited the papers by Lastella, Roach, Hoshikawa and added the paper by Saw (with swimmers). Therefore, we modified the sentence in L60-62 accordignly.

L69-71 : just for my information, why you can argue that they are olympic level open-water swimmers ? Not really important to state that on the manuscript but it is important to mention us how you can say that.

A: Thanks for asking. They participated in the last edition of the Olympic Games. However, we added the reference of the recent paper from McKay et al 2021: Defining Training and Performance Caliber: A Participant  Classification Framework  IJSPP and we classified the participants as Tier 4 and Tier 5

L100-101 : does the fact that the measurements were taken at the beginning (T1) and then at the end of the week (T2 & T3) has an impact of the results ?

A: Thank you for this comment. We think that the timing of the measurements did not impact the results. We monitored all the single nights but decided to report data only from those nights that were in correspondence to the single training blocks.

L126-127 : could you explain why RPE value of 4 is included in Z2 whereas itw as included in Z1 on this article : Baldassarre R, Bonifazi M, Meeusen R, Piacentini MF. The road to Rio: A brief report of training-load distribution of open-382 water swimmers during the Olympic season. Int J Sports Physiol Perform. 2019;14(2).

A: When we published the Baldassarre 2019 paper we utilized the breakpoints used by Seiler (2009) and universally utilized in defining training intensity distribution in endurance athletes. However, moving forward on testing and evaluating open water swimmers we realized that a value of “4” was already beyond the first ventilatory threshold for swimmers as reported by IENO et al 2020 (Ieno C, Baldassarre R, Quagliarotti C, Bonifazi M, Piacentini MF. Session RPE Breakpoints Corresponding to Intensity Thresholds in Elite Open Water Swimmers. J Funct Morphol Kinesiol. 2020;5(1). For this reason, when testing or monitoring this specific population we suggest to utilize the new breakpoints (1-3 in Z1 and 4-6 Z2)

However, considering that it was beyond the scope of this paper to discuss about different monitoring methods, we included sRPE only in the new table requested (and presented as supplementary file) (daily training load) and we did not use it to report training load distribution

L142 : all the results are comparing differences in mean groups (with just 5 subjects). Do you think it is relevant to share some individual changes, or at least to mention if there was individual differences throughout the periods? Please explain here and on the manuscript.

A: Yes, we agree. We believe that analyzing/showing individual changes/values would be relevant in a study with a small sample of subjects but we preferred to maintain the present format of figures/analysis. Actually, we tried to modify the figure showing individual trend lines but the result was messy and confusing for readers. However, the only clear individual changes were observed for sleep onset and offset times (with a clear decrease in T1 for all subjects).

L212-221 : it was not possible to calculate a daily training volume/load in order to also compare data with T1 ? I think the reader to deserve more detailed data for that period.

A: We included a table, presented as supplementary file, with daily volume and sRPE per session.

Figure 2 is clear but could be really improved in terms of graphics. Please delete the grid pattern, the borders. Maybe color could be green for Z1, orange for Z2, and red for Z3.

A:We thank the reviewer for the suggestion. The figure was improved and sRPE was eliminated.

L235-251 : the paragraph is great here but maybe you can shorten some general ideas and add one or two comments about the non-impact on sleep quality/quantity by altitude.

A: Thank you for the positive comment. We decided to maintain that part of the text since it describes the sleep differences dictated by the training schedules. However, we added a sentence at the end on sleep quality and quantity which is perfectly connected to the text in line 265-275.

Is that possible that we did not observe negative impact because of the general poor sleep quantity ?

A: This is an interesting comment. We discussed about the poor sleep quantity observed in our sample of swimmers in lines 245-250. We think that, despite the poor baseline sleep quantity, the ATC at 1500m would have not influenced sleep parameters anyway.

L261 : I think another comment on altitude exposure could be great. I mean the fact that 1500m is moderate altitude and allows to sleep well, but maybe it doesn’t allow sufficient physiological changes ?

A: We agree that an athlete’s sleep at moderate altitude (e.g. 1500 m) is not typically impaired and that sleep disturbances tend to occur only at >2000m. We modified the text accordingly to stress this concept.

L262-272 : I think in this paragraph, two points are missing. The first is some comparison with training intensity distribution in other elite athletes or swimmers. A reference on the polarized training concept could be useful. The second – in line with the previous one – could be to discuss the differences observed on intensity distribution depending on physiological measurements (HR, RPE, volume).

A: We thank the reviewer for this comment. The distribution is more pyramidal than polarized. Considering this was not the focus of the paper (we discussed training intensity distribution in open water swimmers in the Baldassarre et al 2019 paper) I would avoid taking off the focus from the primary outcome which was sleep parameters.

Second, we thought of including a paragraph on the training intensity distribution differences observed with the different measurements but it is beyond the scope of the paper. However, we did add in the discussion a sentence regarding possible differences specifically in altitude between external and internal training load.  

L274-276 : could you tell why swimmers are different ? Or not ?

A: Swimmers, as athletes of individual sport disciplines, can be extremely different (in terms of sleep behavior: quantity, quality and timing) from athletes of team sport disciplines. We cited papers (citations 29-31) to support this concept in the text.